# Evaluation of Grinding of Unfilled and Glass Fiber Reinforced Polyamide 6,6

**DOI:** 10.3390/polym12102288

**Published:** 2020-10-06

**Authors:** Roberto Spina, Bruno Cavalcante

**Affiliations:** 1Dipartimento di Meccanica, Matematica e Management, Politecnico di Bari, 70125 Bari, Italy; bruno.cavalcante@poliba.it; 2Istituto Nazionale di Fisica Nucleare (INFN)—Sezione di Bari, 70125 Bari, Italy; 3Consiglio Nazionale delle Ricerche—Istituto di Fotonica e Nanotecnologie (CNR-IFN), 70126 Bari, Italy

**Keywords:** grinding, material characterization, mechanical testing

## Abstract

This paper investigates the grinding process on unreinforced (PA66) and reinforced glass-fiber polyamide 6,6 (PA66 GF30) with Al_2_O_3_ and SiC abrasive wheels. Both materials were ground by varying rotations, workpiece infeed speed, depth of cuts for sequential roughing/finishing steps. Dry and liquid coolant conditions were also considered during the grinding process to evaluate the effects on part quality. The surface roughness was used to assess the quality of the final products with several parameter combinations, identifying the induced process trends. The results show that at the end of the finishing step, the surface roughness *R_z_* was lower than 4 μm, attaining the lowest value of 1.34 μm for PA66 specimens. The analysis also suggested the choice of the Al_2_O_3_ grinding wheel to reach the lowest *R_z_* values for both materials.

## 1. Introduction

The growing request for environmentally friendly cars forces the automotive companies to explore new potentials and applications of innovative materials. Car manufacturers are effectively employing engineering thermoplastics to balance the constant price increase with the technological advances connected to their use. Patil et al. [1] give an overview of the polymeric material for automotive applications. They pointed out that the natural outcomes of their use are lighter, more fuel-efficient cars, with a larger interior volume, and cheaper replacement parts. The reduction of 5–7% in the amount of fuel consumed is achieved for every 10% reduction in car weight, allowing the whole vehicle weight decline. The use of these lightweight materials introduces new features that keep pace with competitors. For this reason, highly smooth surfaces and tight tolerances are highly required. Kobayashi [2] presents recent results on the ultra-precision machining of thermoplastic to evaluate performances. The idea is to improve machining in substituting the molding process when high form accuracy and surface quality are strictly required. The research confirms that good surface quality can be realized with time-consuming machining, such as turning and milling with feed rates lower than 5 µm/rev, preserving high precision. Wegener et al. [3] suggest grinding as an active process to meet these needs in a shorter time. The authors review the evolution of the grinding process driven by severe demands on accuracy, material removal rate, and surface integrity. In this way, the process ranges from high material-removal rates to finishing and ultra-precision operations, being the final step in the manufacturing process chain. Silva et al. [4] point out that it is crucial, in this context, to evaluate the comparative performance of grinding wheels, evaluating several aspects connected to product specifications and process sustainability. They propose an evaluation methodology that is able to identify the typical curve of the grinding wheel, analyze the output process variables, and define the sustainability indicators. Grinding can also be used for surface structuring to improve tribological properties, with reduced or no supplementary investments in machinery or production steps. As a result, the methodology allows better decision making on the product and wheel efficiency. Rudrapati et al. [5] underline that the grinding process is one of the most complex because it is responsible for the final product quality in terms of geometric accuracy, surface features, and surface roughness. These properties are fundamental due to their influence on certain practical features, such as contact area and characteristics causing surface friction, wearing, light reflection, heat transmission, the ability to distribute and hold a lubricant, load-bearing capacity, and fatigue resistance. Different technical issues appear in processing polymers, making the understanding of the material behavior an important research issue [6]. Thus, a natural settlement is that they rely on thermoplastic material properties. El-Wakil [7] indicates a continuous interest in producing structural parts made of fiber-reinforced thermoplastic composites using precision grinding. However, the author specifies that the components should meet with no-defect grinding operations such as surface degradation and thermal deterioration of the workpiece, excessive machine noise and vibrations during processing, and abrasive wheel clogging to cite a few cases. Moreover, Agarwal [8] investigates the accumulation of chips between grain spaces which may have detrimental effects on processing such as wheel life reduction, higher cutting forces, power requirement, and out-of-tolerance products. These concerns did not happen with metals after the right choice of the process parameters. Besides, defect recovery due to incomplete components, the burr presence, or small technological deviations may appear. The recovery of these non-qualified products using grinding can increment production and trim waste.

Engineering thermoplastics to use in automotive applications should have specific properties such as a low density, outstanding chemical resistance, mechanical strength, and operative temperature up to 120 °C. These features make engineering polymers suitable to the desired applications thanks to a low cost/performance ratio, and competitive to metals in their replacement. However, identifying substitutes is a difficult task if the same functional specifications should be respected. Moreover, the use of thermoplastics as metal replacements can follow a long approval path in the decision framework. Spina et al. [9] demonstrate how difficult the fabrication and acquisition of complex geometry made of thermoplastics are. The authors developed an integrated numerical–experimental framework able to define geometry, simulate the process, and acquire the product geometry, considering the great variation induced by process conditions. This variation is crucial in processing thermoplastics and a successful metal replacement process. We focus our attention on polyamide class, the strength, stiffness, resistance to heat, hydrocarbons, lubricity, and wear, to characterize polyamide (PA)66, as reported by Wypych [10]. The author also points out that PA6 has a better creep strength and glowing surface appearance, whereas modulus, processing temperatures, and mold shrinkage are lower than PA66. However, the moisture absorption of PA6 is higher than that of PA66. The addition of fibers to PA66 allows high-quality surfaces, dimensional accuracy, and surface integrity to be achieved.

The work aims to study the effects of the primary grinding parameters and the abrasive wheel material on workpiece surface quality of PA66 and PA66 GF30 components, examining induced variations and appraising trends. This article extends the experience gained in a previous research on simple flat specimens made of PA66 (Spina and Cavalcante [11]). The article initially describes the materials and methods adopted in this research, in terms of workpiece and grinding wheel materials. Then, the analysis evaluates the use of air and liquid coolant during grinding, respecting the technological process constraints (excessive material softening). Finally, the results of the roughing and finishing passes, starting from machined components, are collected and examined to identify process trends in relation to workpiece material.

## 2. Materials and Methods

A surface of 24 × 20 mm^2^ on ∅ 40 × 20 mm^2^ holed cylinders was realized, as Figure 1 shows. The rough materials were an unreinforced natural PA66, named TECAMID 66, and a black 30% glass-fiber reinforced PA66, named TECAMID 66 GF30, supplied in the form of extruded rods by Ensinger GmbH (Nufringen, Germany). Table 1 reports the nominal properties of the materials.

### 2.1. Machining Process

Machining with a PD 1080 CNC station (Spinner Werkzeugmaschinenfabrik GmbH, Sauerlach, Germany) was initially employed for raw part fabrication using very sharp HSS tools to minimize frictional heat. Figure 2 shows the images of the machined surface acquired with an Evo MA25 Scanning Electronic Microscope (Carl Zeiss AG, Oberkochen, Germany).

The surface of PA66 was regular because the low processing temperatures did not promote the creation of a partially melted layer. Temperatures above the glass transition temperature had an essential effect on microstructural changes and post-shrinkage after cooling. However, machining traces were still visible. The machining traces were less evident than those left on PA66. Davim et al. [12] underline that these fibers are very stiff and strong. Their introduction into the matrix induces directionality or anisotropy due to the type, alignment, distribution, interface, size, and shape of the fiber. In this case, the fibers were mostly oriented along the longitudinal direction. The fibers were uniformly distributed because the glass fibers rotated and aligned according to the extrusion flow, resulting in a homogeneous fiber orientation state in the raw part.

### 2.2. Setup of the Grinding Process

Grinding was then selected to realize the final samples. A Planomat HP 408 machine (Blohm Maschinenbau GmbH, Hamburg, Germany) was used, the features of which were a maximum rotating speed of 1450 rpm and a 7.5 kW maximum spindle power. This CNC machine, supplied initially with one wheel, was customized to operate with two wheels. The first wheel was used for the roughing pass, whereas the second one for the following finishing passes. This CNC machine was chosen due to its large housing volume, which allowed a camera to record machining phases and a thermo-camera to acquire temperature fields to be installed. The workpiece handling was realized with a 3-axis system with the option of rotating the part around its z-axis for precise positioning. The workpiece mounting was carried out using a tailstock.

The investigation was limited on the analysis of some distinct grinding parameters such as the cutting speed *v_c_*, infeed speeds *v_w_*, and depth of cut *a_e_*. The abrasive wheel was also changed to evaluate its influence on the final properties of the part. This investigation was functional for future implementation of an automatic off-line fuzzy system to optimize ground product quality, according to a previous experience of Galantucci et al. [13], able to relate the roughness values with the grinding and dressing parameters. Figure 3 shows the grinding parameters, variation ranges of which are reported in Table 2 in terms of maximum, average, and minimum values.

A rough pass and a consecutive finished pass were carried out. Attention was paid to the position of the component to observe the extrusion direction and, thus, the fiber direction. Three samples were realized for each pass with the same process parameters, measuring the average surface roughness at the end of each grinding step. An aluminum oxide (Al_2_O_3_) and a silicon carbide (SiC) wheel were used, with ∅ 406 mm × 30 mm dimensions. Al_2_O_3_ abrasive was cheap, versatile, and utilized in all grinding works. The fused Al_2_O_3_, combined with the vitrified ceramic matrix, exhibited excellent sharpness characteristics and enhanced wear resistances, as Kopac and Krajnik [14] report.

The SiC wheel, characterized by a needle-shaped abrasive grit, was a little bit more expensive and harder. It was primarily selected in application with parts made of high alloy steels, cast iron, non-ferrous metals, as Bhowmik and Naik [15] confirm. The grinding process is susceptible to the stochastic properties of the abrasive wheel, such as the grain shape and distribution within the matrix, and the matrix chemical and physical properties, citing Guerrini et al. [16]. Coarse grains and an open structure characterized the roughing wheels of both materials, allowing a high material removal volume to be achieved without tool kneading occurred. The finishing Al_2_O_3_ grinding wheel had a smaller grain size and a more closed structure. Figure 4 shows the images of the different grinding wheels acquired with SEM.

The mean roughness depth *R_z_* was the quality parameter used for the process evaluation. According to the ISO 4287:1997 standard, it is defined as the arithmetic mean value of five samples of the absolute vertical distance between the maximum profile peak height and the maximum profile valley depth along sampling length. The *R_z_* value is a critical micro-geometry feature to verify the presence of protruding peaks, which could affect static or sliding contacts. The choice of this parameter, among others, was made considering that the component is in contact when in service with other components, influencing their contact and slicing effects. A scanning speed equal to 0.5 mm/s for a 5.6 mm sampling length was set for the *R_z_* measurement by using a MarSurf XCR20 (Mahr GmbH, Göttingen, Germany). This instrument had a resolution of 0.19 µm using the 175 mm probe arm and 0.04 µm relative to the measuring system.

## 3. Results

### 3.1. Preliminary Analysis with Air-Cooling

Some preliminary tests with the highest values of the process parameters of the roughing pass and the Al_2_O_3_ grinding wheel were performed to evaluate the possibility of working in dry conditions, without using lubrication. Several advantages could be attained with dry processes such as the absence of media residues on components, ruled out of the influence of cooling lubricants on the material, and absence of interactions with the raw material. Moreover, the ability to altogether remove lubricant was an essential step towards a truly sustainable supply chain. Drazumeric et al. [17] suggest analyzing the surface temperature distribution along the grinded profile to select the correct feed increments for improving production rates while avoiding thermal damage. Using this method, grinding with the absence of burns was achieved, avoiding the long consuming trial-and-error approach. Moreover, Awale et al. [18] point out that a large amount of cutting fluid is normally used between wheel and workpiece to preserve the surface integrity, generating several undesirable effects related to operator health, water and soil pollution. In this experiment, a PI 160 infrared thermo-camera (Optris Infrared Sensing LLC, Portsmouth, NH, USA) with an optical resolution of 160 × 120 pixels, a maximum frame rate of 120 Hz and accuracy equal to ±2 °C was employed to acquire the thermal field during dry processing.

The air-cooling condition varied from no flow (0% of the maximum flow rate) to full flow (100% of the maximum flow rate). Figure 5 shows the thermal fields acquired with the thermo-camera during processing. The maximum temperature reached between the tool and part surfaces was monitored using a virtual probe, and values reported in the same figures.

The maximum temperature was always lower for PA66 than PA66 GF30, due to the improved wear resistance of the glass fibers embedded in the PA66 matrix and the resulting polymeric material restriction between the glass fibers, according to Kim et al. [19]. An increase in the flow rate determined a reduction in the maximum temperature. However, these temperatures were too high, causing burns on the machined parts and kneading on the grinding wheel. These results were confirmed by the study of Aurich et al. [20], who focus their attention on the complex relations existing between system, process, and results, accurately balancing these functional attributes. The authors, investigating the process using a numerical model calibrated with the experimental results, report that a higher workpiece temperature caused a workpiece softening, making the process less effective. This result was also confined from the research of Souza Ruzzi et al. [21], who pointed out that the mechanical energy to form chips during grinding is normally transformed into thermal energy, with a ratio between 60 and 90% of the total processing energy.

### 3.2. Liquid Coolant

Based on the above results, the problems related to the high processing temperature were eliminated by adopting the liquid coolant CIMTECH A31F (Cimcool Industrial Products B.V., Vlaardingen, NL, USA) and performing an appropriate dressing before use. The roughing pass and the following finishing pass with liquid coolant for all combinations of part materials and grinding wheels were investigated. No media residues on components and absence of interactions with the raw material were detected on final parts. The maximum temperature reached between the tool and part surfaces remained below the 40 °C for all conditions. The interactions between processing parameters and mean roughness depth *R_z_* were studied. The initial *R_z_* values of as-machined PA66 and PA66 GF30 parts were 50.25 and 48.34 µm (standard deviation *σ* equal to 5.18 and 4.22 µm), respectively. Hu and Zhang [22] investigated the grinding performance of reinforced composites, pointing out that the influence of the fiber orientation was prominent on the surface roughness of ground specimens as well as the grinding force. For this reason, the grinding direction was kept constant for all tests, following the main extrusion direction of the specimens.

The *R_z_* measurements after the roughing pass are shown in Figure 6 and Figure 7. A low depth of cut generated a smooth surface for all cutting speeds and wheel materials. *R_z_* values of all specimens were greater than 5 µm. In particular, the minimum *R_z_* values of 7.21 µm (*σ* equal to 0.71 µm) and 5.20 µm (*σ* equal to 0.44 µm) were achieved with the Al_2_O_3_ grinding wheel on PA66 and PA66 GF30 specimens, respectively. The minimum *R_z_* values obtained with the SiC grinding wheel on PA66 and PA66 GF30 specimens were, respectively, 6.04 µm (*σ* equal to 0.47 µm) and 6.53 µm (*σ* equal to 0.34 µm). These minimum *R_z_* values are indicated with the red arrows in the graphs.

The analysis of these results pointed out that *R_z_* normally grew with *v_w_*. An increase in *v_w_* produced an increase in chip thickness and length, causing an intensification in machining forces and thermal stresses. A considerable contact length coupled to a large infeed produced high grinding forces, leading to machine displacements and, consequently, possible product out of the tolerance dimensions. A reduction in the machining forces was attained with a *v_w_* decrease to minimize the continuous contact grains/workpiece during the process. As a result, chips with smaller thicknesses were attained. Moreover, an increase in *v_w_* accentuated the decay rate of the workpiece surface quality leading to higher *R_z_* values. Higher *a_e_* always produced a very rough surface, with relative detection of the surface quality, independently from the grinding wheel used. A *v_c_* increase made the process unstable due to the influence of the other process parameters and wheel materials. As a consequence, the variation of all parameters, coupled to higher *a_e_*, seemed to make the process unstable. A possible reason could be that a greater friability and loading of the abrasive grains were promoted by a concurrent increase in *a_e_* and *v_w_*, while a *v_c_* increase generated a reverse outcome. This phenomenon was well detected using the Al_2_O_3_ wheel. In fact, a higher *v_c_* coupled with a lower *v_w_* led to smoother surfaces. However, the roughness sharply decreased with a *v_w_* increase and an *a_e_* increase.

The finishing pass was thus needed to further improve the surface quality. Figure 8 and Figure 9 show results using Al_2_O_3_ and SiC wheels, respectively. The values of *R_z_* were lower than those achieved with the roughing pass in all processing conditions. Low *a_e_* values allowed smoother surfaces to be realized.

The Al_2_O_3_ wheel worked better than the SiC wheel because it produced lower forces and contributed to a better surface finish. The lowest *R_z_* values with the Al_2_O_3_ and SiC grinding wheels were, respectively, 1.34 µm (*σ* equal to 0.08 µm) and 3.01 µm (*σ* equal to 0.02 µm) for PA66. The lowest *R_z_* values with the Al_2_O_3_ and SiC grinding wheels for PA66 GF30 were, respectively, 2.53 µm (*σ* equal to 0.07 µm) and 3.99 µm (*σ* equal to 0.03 µm). These values were higher than those measured for the PA66. For lower *v_w_*, low dispersion of *R_z_* was recorded, also with an increase in *a_e_* and *v_c_*. The *R_z_* values of PA66 increased for all other combinations for the PA66 while remaining quite stable for the PA66 GF30. For higher *v_w_* values, the surface quality of PA66 rapidly worsened with an *a_e_* and *v_c_* increase, especially for higher values. The surface roughness of PA66 GF30 seemed to be less sensitive to these variations and not necessarily influenced by *a_e_* within the tested range. With regard to PA66, the processing conditions determined higher forces, producing higher surface roughness. Reduced surface roughness was usually associated with less fracture and strain generation. A lower *a_e_* generated lower cutting forces, with lower vibrations and a surface finish improvement. On the contrary, an *a_e_* increase promoted the heat generation and tool wear, causing a higher surface roughness.

## 4. Conclusions

This study has focused on the analysis of the mean roughness depth of unreinforced (PA66) and reinforced polyamide 6,6 (PA66 GF30) with Al_2_O_3_ and SiC abrasive wheels, performing sequential roughing/finishing steps. Some new considerations were identified from the analysis of the results, in terms of the mean roughness depth *R_z_*. In particular, Al_2_O_3_ has generated coarser surfaces in the rough pass and smoother surfaces in the finishing step on both materials, in comparison with the SiC wheel. The machining parameters were more comfortable to be tuned with the Al_2_O_3_ wheel in the selected ranges. On the contrary, a higher friability of the SiC abrasive led to a high *R_z_* fluctuation. Good results have been achieved by using a low depth of cut coupled to low workpiece infeed and cutting speeds. As concerns the materials, the PA66GF30 has a worsened quality surface due to the fiber presence.

Future investigations will include the measurement of temperature profile and exchanged forces between workpiece and wheel to better understand process phenomena. Moreover, a better understanding of interactions between rubbing, plowing, and chip removal during grinding for these materials is necessary to evaluate the product quality. In this way, the right choice of parameters for the corresponding material will ensure outstanding surface quality with a slight roughness to be attained.

## Figures and Tables

**Figure 1 polymers-12-02288-f001:**
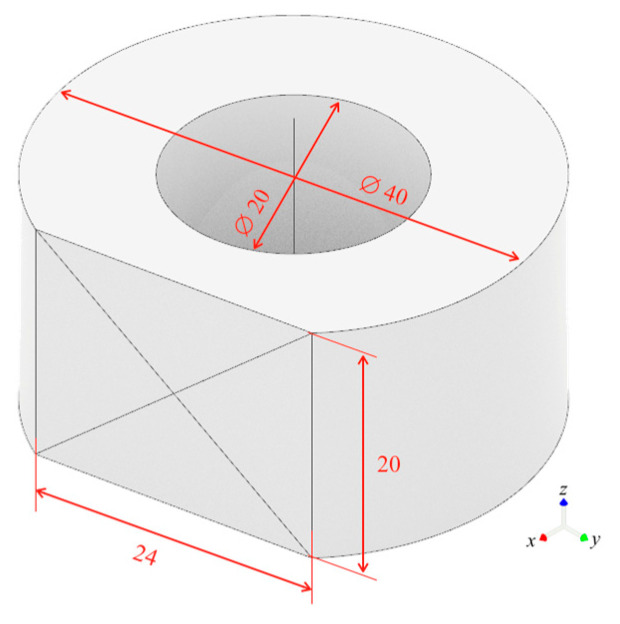
Manufactured raw part (all dimensions in mm).

**Figure 2 polymers-12-02288-f002:**
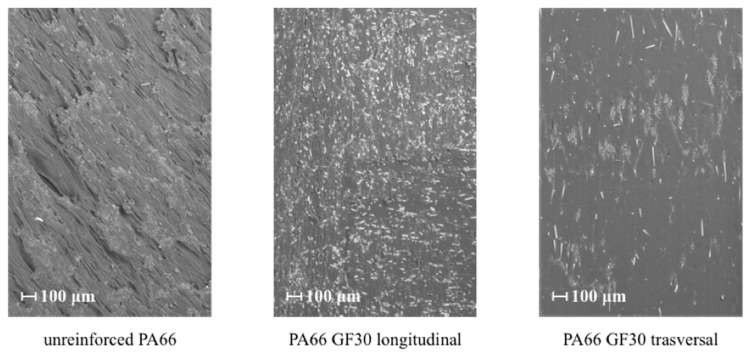
SEM images of the polyamide (PA)66 and PA66 GF30 part surfaces.

**Figure 3 polymers-12-02288-f003:**
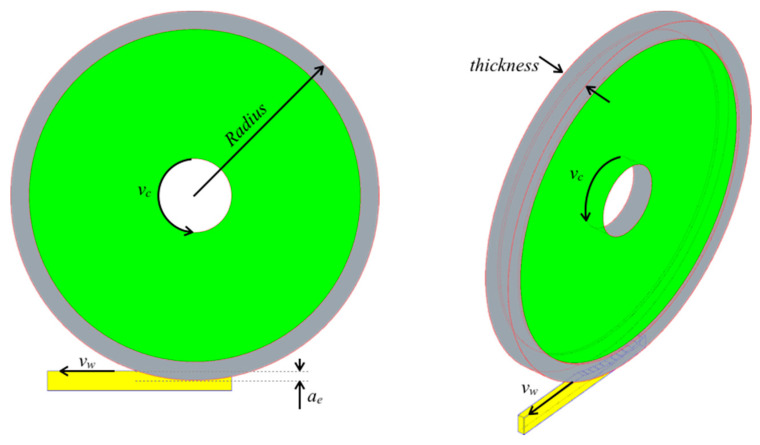
Process parameters.

**Figure 4 polymers-12-02288-f004:**
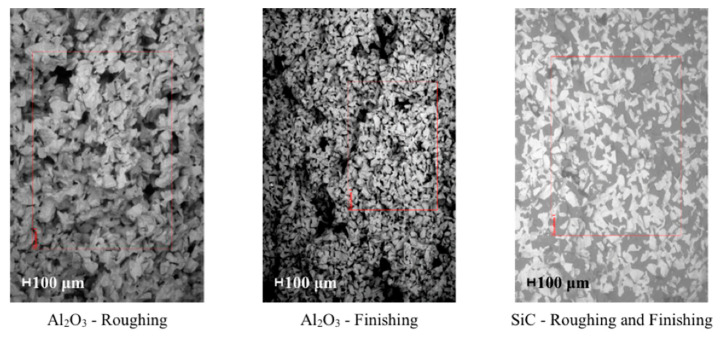
Images of the grinding wheel surfaces with SEM (red rectangles for quantitative analysis).

**Figure 5 polymers-12-02288-f005:**
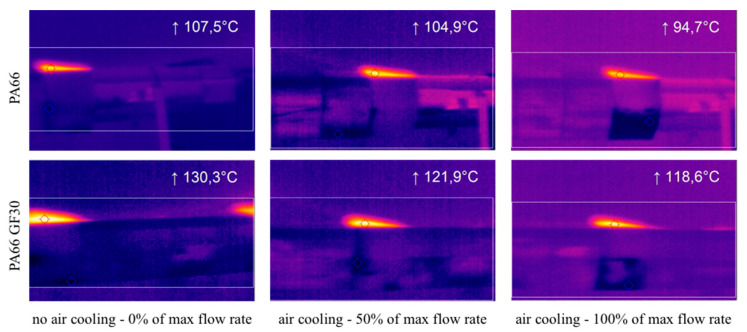
Acquired infrared images during preliminary tests.

**Figure 6 polymers-12-02288-f006:**
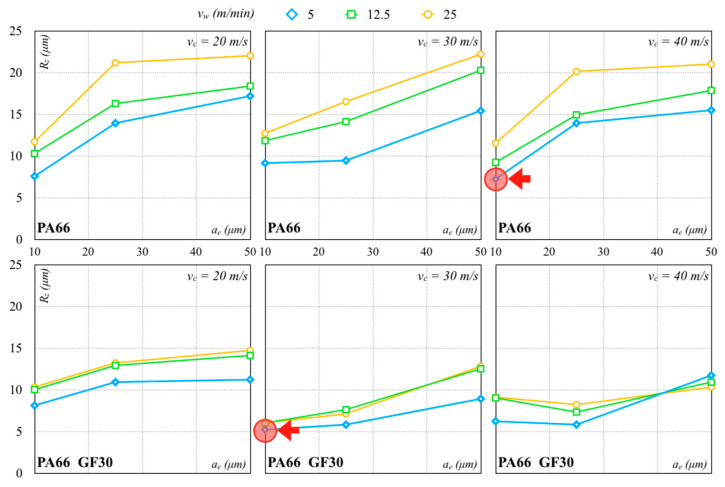
Roughing pass with Al_2_O_3_ abrasive (arrows indicate the minimum values).

**Figure 7 polymers-12-02288-f007:**
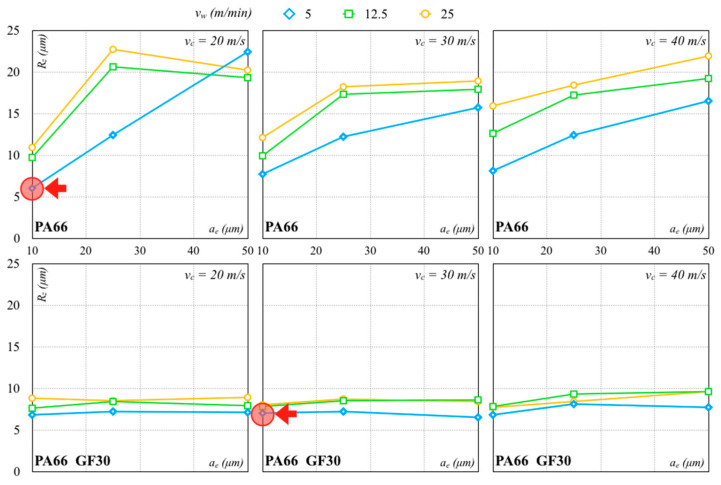
Roughing pass with SiC abrasive (arrows indicate the minimum values).

**Figure 8 polymers-12-02288-f008:**
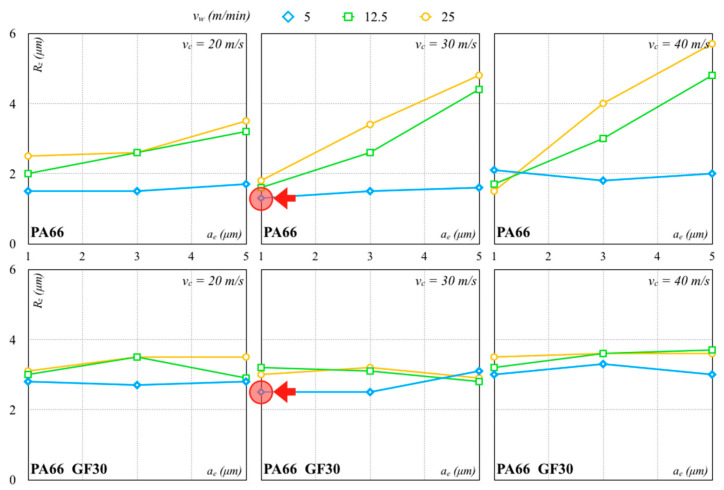
Finishing pass with Al_2_O_3_ abrasive (arrows indicate the minimum values).

**Figure 9 polymers-12-02288-f009:**
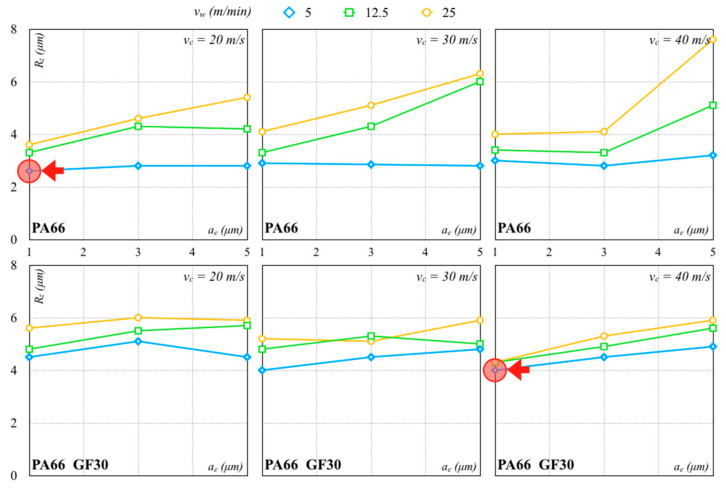
Finishing pass with SiC abrasive (arrows indicate the minimum values).

**Table 1 polymers-12-02288-t001:** TECAMID material properties.

	PA66	PA66 GF30	
Properties	Value		Unit
***Physical***			
*Density*	1.15	1.34	g/cm^3^
*Glass Transition Temperature*	47	48	°C
*Melting (Softening) Temperature*	258	254	°C
***Mechanical***			
*Young’s Modulus*	3.5	5.5	GPa
*Yield Tensile Strength*	84	91	MPa
*Ultimate Tensile Strength*	85	91	MPa
*Elongation at break*	70	14	%
*Coefficient of Thermal Expansion*	120	50	μm/m/°C

**Table 2 polymers-12-02288-t002:** Process parameters.

Process Parameters	Min	Avg	Max	Unit
***Roughing***				
*cutting speed v_c_*	20.0	30.0	40.0	m/s
*infeed speed v_w_*	5.0	12.5	25.0	m/min
*depth of cut a_e_*	10.0	25.0	50.0	μm
***Finishing***				
*cutting speed v_c_*	20.0	30.0	40.0	m/s
*infeed speed v_w_*	5.0	12.5	25.0	m/min
*depth of cut a_e_*	1.0	3.0	5.0	μm

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
