# Peer review of "Evaluation of Grinding of Unfilled and Glass Fiber Reinforced Polyamide 6,6"

_polymers, 2020, doi:10.3390/polym12102288_

Round 1
Reviewer 1 Report
the authors stated ' The objective of this investigation was a basic implementation of an automatic system to examine ground product quality'. no clear indication as to what automatic system used.
relying on Rz alone to quantify the surface is not sufficient, there are nearly 50 parameters to quantify the surface. look at roughness parameters papers.
the work is good but its all been done several times before and to much more in depth analyses.
Author Response
Thank you to the reviewer for the precious suggestions in correcting the paper. We made the necessary modifications.
Q1. the authors stated ' The objective of this investigation was a basic implementation of an automatic system to examine ground product quality'. no clear indication as to what automatic system used.
R1. The sentence was modified to point out to avoid misunderstanding. This investigation was functional for future implementation of an automatic off-line fuzzy system to optimize ground product quality, according to a previous experience of Galantucci et al. [12], able to relate the roughness values with the grinding and dressing parameters. The modification was added to the text.
Q2. relying on Rz alone to quantify the surface is not sufficient, there are nearly 50 parameters to quantify the surface. look at roughness parameter papers.
R2. The Rz (maximum height of the profile) is frequently used to check whether the profile has protruding peaks that might affect static or sliding contact function. The choice of this parameter was done considering that the component is in contact with other components, influencing their contact and slicing effects. The modification was added to the text.
Reviewer 2 Report
The authors compare different ways to fabricate (grind) polymeric parts with a high precission. The work is interesting and overall well described. I only have few comments.
In figure 4, what are the red rectangles in the SEM pictures?
L174-176: such experimental details should be in the experimental
L197 Which liquid coolant? I expected to find this in the experimental, but I could not find details. So tell which coolant has been used (and all the details needed for someone to repeat your experiment). Even more important is to give some proof of the statement before making it (for instance with temperature profiles)! Why this is not mentioned in the conclusions?
Author Response
Thank you to the reviewer for the precious suggestions in correcting the paper. We made the necessary modifications.
Q1. In figure 4, what are the red rectangles in the SEM pictures?.
R1. The red rectangle in the picture was the area for the XDS quantitative analysis. This data was not useful for the paper and was not added to the text.
Q2. L174-176: such experimental details should be in the experimental.
R2. The information was added to the text.
Q3. L197 Which liquid coolant? I expected to find this in the experimental, but I could not find details. So tell which coolant has been used (and all the details needed for someone to repeat your experiment).
R3. The coolant is a CIMTECH A31F, supplied from Cimcool Industrial Products B.V., Vlaardingen, NL. The information was added to the text.
Q4. Even more important is to give some proof of the statement before making it (for instance with temperature profiles)! Why this is not mentioned in the conclusions?
R4. The maximum temperature reached between the tool and part surfaces remained below the 40°C for all conditions. This information was added to the text. To be more precise, this measure cannot be performed with the thermocamera because of the presence of the coolant. The measure was done with a thermocouple positioned 1mm below the top surface of the specimen. However, the description of this measuring system and related results are actually under development.
Reviewer 3 Report
The topic is interesting, and the overall discussion and analysis is in-depth. It deserves to be published after minor revision. A few questions should be addressed:
(1) The SEM pictures were displayed. Are there any magnified morphologies, displaying the effect of worn materials in comparison with un-worn counterparts?
(2) Are there any theoretical analysis of grind machining and the materials?
Author Response
Thank you to the reviewer for the precious suggestions in correcting the paper. We made the necessary modifications.
Q1. In The SEM pictures were displayed. Are there any magnified morphologies, displaying the effect of worn materials in comparison with un-worn counterparts
R1. At the moment, these pictures are under investigations and not added to the present manuscript. Moreover, a better understanding of interactions between rubbing, plowing, and chip removal during grinding for these materials is necessary to evaluate the product quality. The information was added to the text in conclusions as future work.
Q2. Are there any theoretical analysis of grind machining and the materials?
R1. Theoretical studies are limited to the grinding of metals, with very few applications to thermoplastics. .
Round 2
Reviewer 1 Report
No comments